

# Deriving column-integrated thermospheric temperature with the N₂ Lyman–Birge–Hopfield (2,0) band

Clayton Cantrall[1], Tomoko Matsuo[1]

[1]Ann and H.J. Smead Department of Aerospace Engineering Sciences, University of Colorado Boulder, Boulder, CO, USA

*Correspondence to:* Clayton Cantrall (clayton.cantrall@colorado.edu)

**Abstract.** This paper presents a new technique to derive thermospheric temperature from space-based disk observations of far ultraviolet airglow. The technique, guided by findings from principal component analysis of synthetic daytime LBH disk emissions, uses a ratio of the emissions in two spectral channels that together span the Lyman–Birge–Hopfield (LBH) (2,0) band to determine the change in band shape with respect to a change in the rotational temperature of $N_2$. The benefits of the two-channel ratio approach include an elimination of representativeness error as absolute LBH intensities are not required in the derivation procedure and a reduced impact of systematic measurement error caused by variations in the instrumental performance across the LBH band system as a fully resolved system is also not required. It is shown that the derived temperature should, in general, be interpreted as a column-integrated property as opposed to a temperature at a specified altitude without utilization of a priori information of the thermospheric temperature profile. The two-channel ratio approach is demonstrated using NASA GOLD Level 1C disk emission data for the period of 2–8 November 2018 during which a small geomagnetic storm has occurred. Due to the lack of independent thermospheric temperature observations, the efficacy of the approach is validated through comparisons of the column-integrated temperature derived from GOLD Level 1C data with version 2 of the GOLD Level 2 temperature product as well as temperatures from first principle and empirical models. The storm-time thermospheric response manifested in the column-integrated temperature is also shown to corroborate well with hemispherically integrated Joule heating rates, ESA SWARM mass density at 460 km, and GOLD Level 2 column $O/N_2$ ratio.

## 1 Introduction

Remote sensing of Earth's far ultraviolet (FUV) airglow from space provides important insights into the energetics, dynamics, and composition of the upper atmosphere (Meier et al., 1991; Paxton et al., 2017). The $N_2$ Lyman-Birge-Hopfield (LBH) bands (~127-280 nm) are prominent daytime FUV airglow features that emanate from the lower to middle thermosphere (~120-200 km) upon the deposition of solar extreme ultraviolet radiation. Currently operating instruments measuring the LBH bands include the Thermosphere-Ionosphere-Mesosphere Energetics and Dynamics (TIMED) satellite's Global Ultraviolet Imager (GUVI) launched in 2001 (Christensen et al., 2003), the Defense Meteorological Satellite Program (DMSP) Special Sensor Ultraviolet Spectrographic Imager (SSUSI) launched in 2003 (Paxton et al., 2002), the Global-scale Observations of the Limb and Disk (GOLD) launched in 2018 (McClintock et al., 2020a), and the Ionospheric Connection Explorer's Far UltraViolet imaging spectrograph (FUV) launched in 2019 (Mende et al., 2017).

The utility of the LBH bands for probing thermospheric temperature was demonstrated by Aksnes et al. (2006) with limb observations by the Advanced Research and Global Observation Satellite's (ARGOS) High



Resolution Ionospheric and Thermospheric Spectrograph (HITS) instrument. Eastes et al. (2008) subsequently

showed that disk observations of LBH bands could be used for global monitoring of thermospheric temperature. These authors fit LBH laboratory spectra to observed emissions using an optimal estimation routine with varying parameters such as the $N_2$ rotational temperature, population rates of each vibrational band, N 149.3 nm emission intensity, $O_2$ photoabsorption, and background emission rates. GOLD became the first mission to provide a Level 2 data product of thermospheric temperature ($T_{DISK}$) attributed to around 160 km using LBH disk emissions between

~132-162 nm with a similar retrieval implementation (Eastes et al., 2017). Thermospheric temperatures attributed to around 155 km have also been derived from TIMED GUVI observations (Zhang et al., 2019) using an intensity ratio between the (0,0) band and (1,0) band that the authors found to be quasi-linearly dependent on the $N_2$ rotational temperature.

         This paper presents a new technique to derive thermospheric temperature from FUV spectrographic

images. Section 2 provides background and exploration of the LBH temperature signal to motivate the new technique. The technique, unlike in past work, uses the ratio of two spectral channels that span a single LBH band to determine the change in band shape with respect to a change in the rotational temperature of $N_2$. The technique has a straightforward implementation that is relatively unaffected by changes in instrument performance across the LBH band system and that does not require a forward model to reproduce the absolute

intensity of emissions as part of the temperature derivation procedure. These points are further elaborated in Section 3 along with a description of the technique. Section 3 also provides a discussion on potential error sources in the temperature derivation, and a rationale behind our interpretation of the derived temperature as a column-integrated property that we refer to as column-integrated thermospheric temperature, $T_{ci}$. Section 4 presents the demonstrative results of applying the technique to GOLD Level 1C radiance data for the period of 2–8 November 2018; during

which a small geomagnetic storm event has occurred. The derived thermospheric temperatures are compared to GOLD Level 2 $T_{DISK}$ data product over the same period. Due to the lack of independent remotely sensed or in situ temperature measurements in the lower to middle thermosphere, the derived column-integrated temperatures are also compared to (1) synthetically generated column-integrated temperatures from model simulations by NOAA's Whole Atmosphere Model (WAM) (Akmaev et al., 2011), (2) Naval Research Lab Mass Spectrometer and

Incoherent Scatter Radar Extended (NRLMSISE-00) empirical model temperatures (Picone et al., 2002), and (3) observations of other thermospheric states, including GOLD Level 2 $\Sigma O/N_2$ data product (Correira et al., 2018) and mass density by ESA's SWARM constellation (Astafyeva et al., 2017) as well as hemispherically integrated Joule heating rates estimated from SuperDARN and ground-based magnetometer data by using the Assimilative Mapping of Geospace Observations (AMGeO) (AMGeO Collaboration, 2019; Matsuo, 2020).


**2 Temperature Signal in LBH Emissions**

         The thermospheric temperature signal is present in the rotational structure of the LBH bands. In the case of $N_2$, the rotational temperature is equivalent to the ambient neutral temperature (Aksnes et al., 2006). In order to motivate the new technique to extract this signal from the $N_2$ LBH (2,0) band, this section presents results from

principal component analysis (PCA) performed on simulated LBH emissions. It is shown that PCA analysis helps



quantify the amplitude and signal-to-noise ratio (SNR) of this temperature signal in disk LBH emission data. Synthetic LBH emission data are generated by applying forward modeling explained in Section 2.1 to WAM simulation results for the period of 2–8 November 2018. WAM simulation experiments are executed with solar and geomagnetic forcing conditions specified according to the actual values of the F10.7, Kp, and hemispheric power indices, solar wind velocities and densities, and interplanetary magnetic fields. Section 2.1 discusses forward modeling of LBH emissions and Section 2.2 presents the PCA results.

### 2.1 Forward Modeling of LBH Emissions

The forward model used to produce synthetic LBH emissions is built with the Global Airglow Model (GLOW) and a radiative transfer model (Solomon, 2017). GLOW computes LBH volume emission rates as a function of altitude that are input into the radiative transfer model to produce line-of-sight emissions of the LBH band system. The two primary mechanisms generating LBH emissions are direct excitation from the $N_2$ ground state by photoelectron impact and radiative and collisional cascading between the singlet states (Eastes et al., 2000a). GLOW currently does not consider cascading effects for the LBH vibrational states (Solomon, 2017). Eastes et al. (2000b) showed that cascading increases the total LBH emission by a factor of ~1.6. The primary extinction source of LBH emissions is $O_2$ photoabsorption, computed line-by-line in the radiative transfer model.

The most important component of the forward model for the purposes of deriving thermospheric temperatures is the LBH vibrational-rotational band model (Budzien et al., 2001). The band model is essentially a look-up table of laboratory LBH spectra that specifies, for a given temperature, a unique spectrum for the upper vibrational states v'=0–9 of $N_2$. In the current implementation of the forward model, the v'=0–9 vibrational population rates are defined by a Franck–Condon based theoretical model and are held constant (Ajello et al., 1985; Ajello et al., 2020). Ajello et al. (1985) states that significant deviations from a Franck–Condon factor distribution can occur in planetary dayglow or aurora due to the energy distribution of the electron flux, and that excitation thresholding should be included in airglow models to accurately reproduce LBH band intensity. However, as discussed in the following section, absolute band intensity is not needed to extract the $N_2$ rotational temperature.

### 2.2 PCA of Simulated LBH Emissions

PCA is a data reduction technique that is useful for identifying the dominant orthogonal modes of variability from data. PCA is applied here using eigenvalue decomposition on aggregated data sets of simulated emissions of the LBH band system, $I^s_{LBH}$, during 2–8 November 2018 for a total of $8.1 \times 10^4$ samples. The useful results of PCA for this investigation are a set of eigenvectors (principal components), $\mathbf{v}$, that describe the mode of variability with associated eigenvalues, $\sigma$. Suppose that $\mathbf{v}$ is time-invariant and time-dependent coefficients, $\mathbf{c}$, represent the amplitude of the mode for each disk emission sample at a given time, where the total variance of $\mathbf{c}$ matches $\sigma^2$ for that mode.

Figure 1 shows the mean of simulated LBH radiance, $\overline{I^s_{LBH}}$, between 138–162 nm with a spectral sampling of 0.04 nm along with the first two leading modes of variability in the spectrum, $\mathbf{v}_B$ and $\mathbf{v}_T$, scaled by their eigenvalues or total standard deviations, $\sigma_B$ and $\sigma_T$. The leading mode $\mathbf{v}_B$ is identified as the overall scaling of the



LBH intensity due to changes in solar and geomagnetic conditions, solar zenith angle (SZA), and emission angle. The value of $\sigma_B{}^2$ suggests that this mode accounts for 98.3% of the total variability in the simulated LBH spectra.

The second leading mode $\boldsymbol{v}_T$ is identified as the temperature signal (as explained later). According to the value of $\sigma_T{}^2$ this secondary mode accounts for 1.6% of the total variability in the simulated LBH spectra and is just 3% of the intensity of an LBH band. The correlation coefficient, $R$, between time-dependent coefficients for this temperature mode $\boldsymbol{c}_T$ and the simulated WAM temperatures at 160 km altitude over the course of 2–8 November 2018 is 0.68. Together these two principal components account for 99.9% of the variability in the simulated LBH

spectra. Figure 1 highlights that variability of the simulated LBH disk emission spectra can be represented in a small number of degrees of freedom (i.e., there is a significant amount of redundant information) and that the temperature signal is considerably weaker than the overall LBH intensity.

Figure 2 focuses on the LBH (2,0) band from the LBH band system shown in Fig. 1. Compared to the other LBH bands the (2,0) band is relatively bright and is isolated from the even brighter O emissions at 130.4 nm and

135.6 nm and the N emission at 149.3 nm (not shown in Fig. 1). Figure 2 shows $\boldsymbol{v}_T$ scaled by $\sigma_T$ along with the amplitude of shot noise, $\sqrt{I^s_{LBH_{2,0}}}$. The magnitude of temperature mode $\boldsymbol{v}_T$ is less than the level of shot noise (SNR≈0.35), underscoring the need for denoising approaches to detect the temperature signal with sufficient SNR. A simple way to improve SNR, at the cost of spectral or spatial resolution, is with binning.

The temperature signal in LBH emissions manifests as the second leading mode of emission variability as

is apparent in the morphological shape of $\boldsymbol{v}_T$ displayed in Fig. 2. As the rotational temperature of $N_2$ increases there is an effective skewing of the LBH (2,0) band to longer wavelengths. The close inspection of $\boldsymbol{v}_T$ indicates that LBH (2,0) band emissions at wavelengths above 138.58 nm are positively correlated with temperature while those below are negatively correlated. Emissions at 138.58 nm are not affected by temperature variability so have zero amplitude in $\boldsymbol{v}_T$. This observation substantiates an approach of binning the LBH (2,0) band into two channels using 138.58 nm

as a boundary to preserve how the temperature signal manifests in the LBH emission's morphological shape. Channel A is defined as the sum of all wavelengths negatively correlated with temperature ($\sum_{\lambda=138.0}^{138.56} I_\lambda$), and channel B contains all wavelengths positively correlated with temperature ($\sum_{\lambda=138.56}^{139.2} I_\lambda$). The ratio B/A is a linear function of temperature. A similar two-channel ratio approach was adopted in Cantrall et al. (2019) for testing the feasibility of assimilating GOLD Level 1C data into the WAM but a justification of such an approach was not

provided.

In this study, the LBH (2,0) band is sampled at 0.04 nm resulting in 30 bins over 138.0–139.2 nm. Binning these 30 bins into two channels improves the SNR by a factor of about 4 from 0.35 to 1.4. While this SNR is still low, the temperature signal in the LBH (2,0) band is now above the shot noise level. The results from PCA of simulated data of LBH disk emissions suggest that the ratio of the two spectral channels A and B that span a single

LBH band (2,0) as defined above is sufficient to determine the change in the band shape related to temperature variation while improving the SNR of the temperature signal in LBH emissions.

**3 Determination of Column-integrated Temperature from LBH (2,0) Band**





This section presents the procedure to determine column-integrated thermospheric temperature, $T_{ci}$, from the $N_2$

rotational structure observed in top-of-atmosphere LBH emissions using the ratio of two channels that together span the LBH (2,0) band as motivated in Section 2. Section 3.1 explains the procedure step by step, which is followed by a discussion on potential error sources of $T_{ci}$ in Section 3.2 and some analysis in Section 3.3 that supports the interpretation of $T_{ci}$ as a column-integrated temperature rather than a temperature attributed to a specific altitude.

**3.1 Procedure**

The procedure consists of four steps as follows:

1.  Generate synthetic data of spectrally resolved LBH (2,0) band emissions for a range of temperature using the vibrational-rotational band model.

2.  Apply an instrument model on the band model spectra data to calibrate for the instrument's

wavelength resolution and wavelength registration.

3.  Bin the band model spectra data into channels A and B. Tabulate the relationship of B/A ratio to temperature as a look-up table.

4.  Compute B/A ratio on observations of the LBH (2,0) band and determine $T_{ci}$ by linearly regressing observed B/A ratio on the predetermined relationship between B/A ratio and

temperature.

The two-channel ratio approach has a number of benefits, most importantly, it can limit the impact of the following uncertainties: (1) uncertainty associated with LBH excitation and extinction processes that affect the absolute intensity of each band, and (2) uncertainty associated with both the low SNR of the temperature signal and instrument performance variations across the LBH band system (See Section 3.2). In this approach, a measurement

of a fully resolved LBH band system is not required to derive $T_{ci}$ nor is a forward model to produce absolute LBH intensity.

**3.2 Sources of Error**

In general, there are two categories of error associated with determining physical parameters from

observations: measurement error and representativeness error. Measurement error is the error associated with the measuring device while representativeness error is the difference between the observation and the physical model's representation of the observation (Rodgers, 2000). There are measurement errors, both random and systematic, in $T_{ci}$, but the two-channel ratio approach effectively eliminates representativeness error by not requiring a forward model to produce absolute LBH band intensity. However, sources that cause relative differences in the channel

intensity other than temperature must also be quantified. For example, changes in the $O_2$ absorption cross section over the band is the predominant source causing a 1.5% change in the relative intensity.

The predominant source of random measurement error that determines the precision in $T_{ci}$ is shot noise. For the case study with GOLD data, the $T_{ci}$ random measurement error associated with GOLD observations is quantified using Monte Carlo (MC) samples of simulated $T_{ci}$ derivations considering the viewing conditions and instrument





performance (McClintock et al., 2020a,b). An estimate of the errors using MC samples is provided along with case study results in Section 4.1.

Reliance on only a small bandwidth of LBH emissions simplifies uncertainty quantification of $T_{ci}$ derivations due to changes in instrument performance across the entire band system. There are still two dominant sources of systematic bias in $T_{ci}$ stemming from variations in the instrument's wavelength registration and

resolution. Figure 3 shows the bias in $T_{ci}$ as a function of the error in the modeled wavelength registration and the error in the wavelength resolution. It is apparent in Fig. 3 that a significant bias of about 50 K (5–10%) can occur if the errors exceed a hundredth of a nanometer level for the wavelength registration and a tenth of a nanometer level for wavelength resolution. A discussion on how these two sources of biases can be mitigated in deriving $T_{ci}$ from actual GOLD data is provided in Section 4.1.


### 3.3 Interpretation of Column-integrated Temperature

Interpretation of column-integrated temperature, $T_{ci}$, is addressed using synthetic LBH disk emission observations generated by forward modeling WAM simulation results. The column-integrated temperature computed from synthetic observations is hereafter denoted as $T_{ci}^S$ to contrast to $T_{ci}^G$ computed from GOLD LBH disk

emission data that is introduced later. To examine if $T_{ci}^S$ can be attributed to a certain altitude we compare the WAM altitude with a temperature that most closely matches $T_{ci}^S$, denoted as $z_{T_{ci}^S}$, to the altitude at the peak of the LBH contribution function, $z_{\tau=1}$, where altitude where the LBH optical depth, $\tau$, is unity. $z_{T_{ci}^S}$ and $z_{\tau=1}$ are computed over the entire simulation period of 8–13 November 2018 and are shown in Fig. 4 with respect to solar SZA (black). For low SZA, $T_{ci}^S$ is shown to match most to WAM temperatures at altitudes, $z_{T_{ci}^S}$, that are 5–10 km higher (about 30–90

K hotter) than suggested by $z_{\tau=1}$. The gap between $T_{ci}^S$ and temperature at $z_{\tau=1}$ becomes smaller as SZA increases. This dependence of the difference between $z_{T_{ci}^S}$ and $z_{\tau=1}$ on SZA is explained by also considering the full-width-half-max (FWHM) of the contribution function. Figure 4 shows that the contribution function peaks at about 153±2.5 km for SZA = 0° and about 185±10 km for SZA = 75°. The FWHM associated with these contribution functions span about 120–180 km and about 160–250 km for the SZA = 0° and SZA = 75° cases, respectively

(Laskar et al., 2020). In the case of SZA = 0° the FWHM spans an altitude range in the lower thermosphere with the steepest thermospheric temperature gradients resulting in a wide range of temperatures contributing to $T_{ci}^S$ with a net effect of a higher apparent temperature than that at $z_{\tau=1}$. As SZA increases the FWHM spans an altitude range where the thermosphere becomes isothermal in the middle thermosphere resulting in a smaller range of temperatures contributing to $T_{ci}^S$ so the apparent temperature looks like the temperature at $z_{\tau=1}$.

Figure 4 furthermore reinforces that $T_{ci}$ derived from the procedural steps specified in Section 3.1 is indeed a column-integrated quantity, containing information from a larger altitude range of the lower-middle thermosphere than just at $z_{\tau=1}$. Perhaps, $T_{ci}$ can be justified to be attributed to $z_{\tau=1}$ when measurement and representativeness errors exceed the gap between $T_{ci}$ and the temperatures at $z_{\tau=1}$ at a given SZA. In general, specific altitude attribution of $T_{ci}$ requires additional a priori knowledge of the thermospheric temperature profile as demonstrated



with $T_{ci}^s$ derived from synthetic LBH disk emission observations wherein the a priori information is given by WAM simulations.

## 4 Case Study

The two-channel ratio approach to derive the column-integrated temperature is demonstrated using NASA GOLD

Level 1C disk FUV emission data, $T_{ci}^G$, for the period of 2–8 November 2018 during which a small geomagnetic storm has occurred. Due to the lack of independent thermospheric temperature observations, the efficacy of this approach is validated through comparisons with GOLD Level 2 temperature product ($T_{DISK}$) and two-channel column-integrated temperatures computed from the synthetic observations using the WAM simulations ($T_{ci}^s$) as described in Section 2. $T_{ci}^G$ is also compared to NRLMSISE-00 temperatures sampled at the altitude $z_{T_{ci}^s}$ based on the

SZA. This sampled MSIS temperature is denoted as $T_{MSIS}$. $T_{ci}^G$ and $T_{DISK}$ are equivalent variables only differing in the approach to determine column-integrated thermospheric temperature. Note that version 2 of $T_{DISK}$ is used in these comparisons presented in the paper. The approach is further corroborated through comparisons of the storm-time changes of $T_{ci}^G$ to hemispherically integrated Joule heating rates ($Q_{JH}$) estimated from SuperDARN and ground-based magnetometer data using AMGeO, ESA SWARM mass density measurements at 460 km ($\rho_{SWARM}^{460\,km}$) based on

calculation from precise orbit determinations using the Global Positioning System receivers on the spacecraft, and GOLD Level 2 $\Sigma O/N_2$ product ($\Sigma O/N_2^G$). Section 4.1 provides a description of the GOLD LBH Level 1C disk emission data used in the $T_{ci}^G$ derivation, Section 4.2 presents results comparing $T_{ci}^G$ with $T_{DISK}$, $T_{MSIS}$, and $T_{ci}^s$, and Section 4.3 presents results comparing the storm time response of $T_{ci}^G$ with $Q_{JH}$, $\rho_{SWARM}^{460\,km}$, and $\Sigma O/N_2^G$. Table 1 defines each of the variable symbols introduced above.


### 4.1 GOLD LBH Disk Emission Data

GOLD observes the daytime FUV airglow from ~134–162 nm on Earth's disk between 6:10 and 20:10 UTC from geostationary orbit at 47.5°W longitude (Eastes, 2020). GOLD produces a full disk image every ~30 minutes at a spatial resolution of 125×125 km by alternating between scans of the Northern and Southern

hemisphere. The GOLD Level 1C radiance data with a spectral sampling of 0.04 nm are used to derive $T_{ci}^G$ in this study. The GOLD Level 1C data is spatially binned by 2×2 (250×250 km spatial resolution) to improve the SNR by a factor of 2. Prior to deriving $T_{ci}^G$, efforts were made to reduce the impact of systematic biases that are present in version 2 of the GOLD Level 1C data product. Variations in wavelength registration along the GOLD detector are identified with the location of the LBH (2,0) band peak through fitting a log-normal distribution. Variations in

wavelength resolution are identified with the FWHM of the OI 135.6 doublet through fitting a 2-gaussian distribution. Corrections for wavelength registration and resolution are incorporated into Step 2 of the $T_{ci}$ algorithm (see Section 3).

### 4.2 Comparing $T_{ci}^G$ to $T_{DISK}$, $T_{MSIS}$, and $T_{ci}^s$


Figure 5 displays $T_{ci}^G$ along with $T_{DISK}$, $T_{MSIS}$, and $T_{ci}^s$ over Earth's disk viewed by GOLD for a five day window from 3–7 November 2018 at about 15 UTC, noon LT at the center of the disk (47.5ºW, 0ºN). A small geomagnetic storm has commenced the evening of November 4th and lasted through November 5th. $T_{ci}^G$ agrees most with $T_{MSIS}$ in the morphological temperature response to geomagnetic activity over the disk. There is also good agreement in the temperature morphology over the disk between $T_{ci}^G$ and $T_{ci}^s$ prior to the storm, but the storm-time

response simulated by WAM, as manifest in $T_{ci}^s$, shows considerably higher temperatures in the mid- and high-latitudes and a longer post-storm recovery time in comparison to $T_{ci}^G$ as well as $T_{MSIS}$. $T_{ci}^G$ and $T_{DISK}$ show general agreement, particularly at low latitudes, but $T_{DISK}$ exhibits a significant north-south temperature gradient and a relatively constant east-west gradient both of which are absent in $T_{ci}^G$ as well as $T_{MSIS}$. Note that there is slight banding near the equator in $T_{ci}^G$ where the southern and northern hemisphere scans meet that is likely due to

systematic errors (see Section 3.2) at the top and bottom edge of the detector that were not completely corrected.

        Figure 6 shows the root mean squared difference (RMSD) and mean bias difference (MBD) of $T_{ci}^G$ from $T_{DISK}$, $T_{MSIS}$, and $T_{ci}^s$ that is computed for a 5° latitudinal bin. In general, there is agreement of $T_{ci}^G$ with $T_{DISK}$, $T_{MSIS}$, and $T_{ci}^s$ at low latitudes as suggested by the RMSD and MBD. The consistent −25 K MBD at low latitudes for each paired difference, $T_{ci}^G - T_{DISK}$, $T_{ci}^G - T_{MSIS}$, and $T_{ci}^G - T_{ci}^s$, suggests that there is an overall bias in $T_{ci}^G$ likely due to

wavelength registration and/or resolution errors. There are also small-scale fluctuations in MBD observed in each pair that suggests small systematic errors in $T_{ci}^G$.

        As expected from the results shown Fig. 5, there are significant changes in bias as a function of latitude for $T_{ci}^G - T_{DISK}$ (ranging from −50 K to 40 K) and $T_{ci}^G - T_{ci}^s$ (ranging from −145 K to −10 K). The good agreement with $T_{MSIS}$ compared to the other variables is again apparent in Fig. 6 with an RMSD ranging from 60 K to 100 K (6–

18%) that is near the level of the random measurement uncertainty for a certain latitude and a negative temperature bias that gradually increases from 0 K in the southern high latitudes to −45 K (4–8%) in the northern high latitudes. It is not known at this time if this latitudinal trend in bias can be attributed to systematic measurement errors that change along the detector, biased altitude sampling of NRLMSISE-00 temperatures, and/or if NRLMSISE-00 is not completely capturing the latitudinal temperature gradient during this period. Further investigation is required to

reconcile such systematic biases between observations and models. For example, climatological comparisons of $T_{ci}^G$, $T_{DISK}$, $T_{MSIS}$, and $T_{ci}^s$ along with TIMED GUVI derived temperatures over the full GOLD mission lifetime may likely shed light on this issue. This should be performed once version 3 of $T_{DISK}$ is released which is expected to have significant improvements to version 2 used in this paper. Overall, the good agreement with $T_{MSIS}$ over this time period demonstrates the validity in this two-channel ratio approach in probing thermospheric temperature using

LBH disk emissions.

### 4.3 Storm Time Response

        Figure 7 displays the response to a small geomagnetic storm in $T_{ci}^G$, $Q_{JH}$, $\rho_{SWARM}^{460\,km}$, and $\Sigma O/N_2^G$. The storm has started on 4 November 2018 and lasted to 5 November 2018 (Gan et al., 2020). $T_{ci}^G$, $\rho_{SWARM}^{460\,km}$, and $\Sigma O/N_2^G$ are

shown as percent differences from the quiet-time conditions on 2 November 2018. The global temporal evolution of





these variables is in good agreement with each other and consistent with known storm time responses of thermospheric variables (e.g., Fuller–Rowell et al., 1994). A rise of magnetospheric energy influx as suggested by $Q_{JH}$ leads to increased temperatures and upwelling of heavy molecular rich air in the high- and mid-latitudes as indicated by depletions of $\Sigma O/N_2^G$ (–40% near 50º latitude and –20% near –50º latitude) and enhancements of $T_{ci}^G$

(~20% near $\pm 50º$ latitude) and $\rho_{SWARM}^{460\ km}$ (~250% near $\pm 50º$ latitude). Enhancements of $\Sigma O/N_2^G$ (20–30% near 30º latitude) in the low-latitudes suggests a subsequent development of downwelling following the pole to equator global circulation in response to the storm-time Joule heating rise. Global thermospheric expansion is also apparent on November 5$^{th}$ as suggested by an increase of $T_{ci}^G$ and $\rho_{SWARM}^{460\ km}$ over all latitudes. Note that the first detection of the temperature change was on the evening of 4 November when Joule heating rates have started to increase but are still

relatively low (< 50 GW). The post-storm recovery times are also in good agreement and appear to be on the order of 2–3 days.

**5 Conclusions**

A new technique to derive thermospheric temperature from space-based disk observations of FUV airglow

is presented. The technique uses a ratio of the emissions in two spectral channels that together span the Lyman–Birge–Hopfield (LBH) (2,0) band to determine the change in band shape with respect to a change in the rotational temperature of $N_2$. The definition of column-integrated thermospheric temperatures and other parameters used for comparison in the paper is given in Table 1. Specific findings of this work are as follows.

The LBH temperature signal is quantified with PCA of synthetic daytime LBH disk emission data to be just

3% of the intensity of an LBH band and accounts for just 1.6% of the total variability in the simulated LBH disk emissions. The PCA result underscores the difficulty in extracting the thermospheric temperature signal from LBH disk emissions and the necessity of denoising efforts to improve the SNR. Analysis of the secondary principal component mode, that characterizes how the LBH temperature signal manifests as the change in band shape, substantiates the approach to bin a LBH spectral band into two channels such that the temperature-induced band

shape change is best preserved. The SNR of the temperature signal is improved with the two channels compared to a fully resolved band. The study has shown that thermospheric temperatures can be derived from an observed two-channel ratio by using a precomputed relationship of the ratio to temperature from an LBH vibrational-rotational band model. In this two-channel ratio approach, representativeness errors originating from forward modeling are effectively eliminated because absolute LBH band intensities are not required in the derivation procedure, and

negative impact of systematic measurement errors, stemming from variations across the band system in the instrument's wavelength registration and resolution, is reduced because a fully resolved LBH band system is not required.

The derived temperature from the two-channel approach can have significant systematic biases of about 50 K (5–10%) if the wavelength registration and resolution are not known to the hundredth of a nanometer level and

tenth of a nanometer level, respectively, as shown in Fig. 3. In addition to these known sources of systematic biases, there is intrinsic random error in $T_{ci}$ from shot noise. This shot noise is estimated to be 50–95 K (5–15%) for the



case of GOLD L1C data using MC simulations considering the viewing conditions and instrument performance as shown as gray dash line in Fig. 6.

330        The derived temperature is, in general, a column-integrated property referred to as column-integrated thermospheric temperature, $T_{ci}$. $T_{ci}$ should not be attributed to the peak of the LBH contribution function without consideration of SZA and $T_{ci}$ derivation uncertainty. This is particularly true for low SZA, where the observed LBH emissions emanate from a region of sharp temperature gradients. Using a priori knowledge of the thermospheric temperature profile from WAM simulations, as shown in Section 3.2, $T_{ci}^s$ matches most to temperatures at altitudes 5–10 km higher (about 30–90 K hotter) than at $z_{\tau=1}$ for low SZA. As SZA increases, the gap between $T_{ci}^s$ and

temperature at $z_{\tau=1}$ becomes smaller.

       In a case study for the period of 2–8 November 2018 during which a small geomagnetic storm has occurred, $T_{ci}$ agrees with $T_{MSIS}$ across the disk with a bias that spans from 0 to -45 K with latitude and to a lesser degree with GOLD Level 2 temperature product. At low latitudes $T_{ci}^G$ agrees well with $T_{DISK}$, $T_{ci}^s$, and $T_{MSIS}$ with a consistent mean-bias-difference of about –25 K. Mean-bias and root-mean-squared differences between $T_{ci}^G - T_{DISK}$

and $T_{ci}^G - T_{ci}^s$ vary with latitudes as well as geomagnetic activity levels. The temporal evolution of global $T_{ci}$ corroborates well with temporal changes of hemispherically integrated Joule heating rates $Q_{JH}$, SWARM mass density at 460 km $\rho_{SWARM}^{460\,km}$, and GOLD $\Sigma O/N_2^G$, which is consistent with known storm time responses of thermospheric variables.

**Data Availability**

The derived column-integrated temperatures for the period of November 2–8, 2018 using the procedure presented in this paper are available at https://doi.org/10.17605/OSF.IO/KHNQ7. GOLD L1C and L2 data can be accessed at the GOLD Science Data Center (http://gold.cs.ucf.edu/search/) and at NASA's Space Physics Data Facility (https://spdf.gsfc.nasa.gov). The code for NOAA's WAM model is available at https://github.com/NOAA-

SWPC/WAM. The code for the NRLMSISE-00 neutral atmosphere model is available from the NASA CCMC, at ftp://hanna.ccmc.gsfc.nasa.gov/pub/modelweb/atmospheric/msis/nrlmsise00/. The Python interface for the NRLMSISE-00 neutral atmosphere model is available at https://github.com/st-bender/pynrlmsise00. Near-Earth solar wind data is provided by the Goddard Space Flight Center Space Physics Data Facility and is available at https://omniweb.gsfc.nasa.gov/. The density measurements (L2 DNSxPOD data product) from Swarm can be

obtained through the web site at https://earth.esa.int/web/guest/swarm/data-access upon registration. AMGeO is an open source software available from amgeo.colorado.edu upon registration. SuperMAG ground magnetometer data is available at https://supermag.jhuapl.edu/. SuperDarn radar data is available at http://vt.superdarn.org.

**Author Contribution**

CC developed the presented technique and performed the analyses. TM contributed AMGeO, determined the validation approach and provided interpretation of the analyses. CC and TM prepared the manuscript.

**Competing Interests**



The authors declare that they have no conflict of interest.


**Acknowledgements**

The authors acknowledge W.E. McClintock for his assistance with the GOLD data, S. Solomon for his assistance with the use of GLOW model, A. Kubaryk for his assistance with the use of WAM, and L. Kilcommons for his assistance with the use of AMGeO. This work is supported by the NASA Future Investigator in NASA Earth and
Space Sciences (FINESST) award 80NSSC19K1432. TM is supported by the NSF CAREER award AGS-1848544. AMGeO is supported by the NSF EarthCube awards ICER 1928403, ICER 1928327, and ICER 1928358. The authors acknowledge the use of SuperDARN data. SuperDARN is a collection of radars funded by national scientific funding agencies of Australia, Canada, China, France, Italy, Japan, Norway, South Africa, United Kingdom and the United States of America. For SuperMAG data we are grateful for INTERMAGNET, Alan
Thomson; CARISMA, PI Ian Mann; CANMOS, Geomagnetism Unit of the Geological Survey of Canada; The S-RAMP Database, PI K. Yumoto and Dr. K. Shiokawa; The SPIDR database; AARI, PI Oleg Troshichev; The MACCS program, PI M. Engebretson; GIMA; MEASURE, UCLA IGPP and Florida Institute of Technology; SAMBA, PI Eftyhia Zesta; 210 Chain, PI K. Yumoto; SAMNET, PI Farideh Honary; IMAGE, PI Liisa Juusola; Finnish Meteorological Institute, PI Liisa Juusola; Sodankylä Geophysical Observatory, PI Tero
Raita; UiT the Arctic University of Norway, Tromsø Geophysical Observatory, PI Magnar G. Johnsen; GFZ German Research Centre For Geosciences, PI Jürgen Matzka; Institute of Geophysics, Polish Academy of Sciences, PI Anne Neska and Jan Reda; Polar Geophysical Institute, PI Alexander Yahnin and Yarolav Sakharov; Geological Survey of Sweden, PI Gerhard Schwarz; Swedish Institute of Space Physics, PI Masatoshi Yamauchi; AUTUMN, PI Martin Connors; DTU Space, Thom Edwards and PI Anna Willer; South
Pole and McMurdo Magnetometer, PI's Louis J. Lanzarotti and Alan T. Weatherwax; ICESTAR; RAPIDMAG; British Artarctic Survey; McMac, PI Dr. Peter Chi; BGS, PI Dr. Susan Macmillan; Pushkov Institute of Terrestrial Magnetism, Ionosphere and Radio Wave Propagation (IZMIRAN); MFGI, PI B. Heilig; Institute of Geophysics, Polish Academy of Sciences, PI Anne Neska and Jan Reda; University of L'Aquila, PI M. Vellante; BCMT, V. Lesur and A. Chambodut; Data obtained in cooperation with Geoscience Australia, PI
Andrew Lewis; AALPIP, co-Pis Bob Clauer and Michael Hartinger; SuperMAG, PI Jesper W. Gjerloev; Data obtained in cooperation with the Australian Bureau of Meteorology, PI Richard Marshall.

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





**Tables and Figures**


| | |
|---|---|
| $T_{ci}^G$ | Column-integrated thermospheric temperature derived from GOLD L1C disk data |
| $T_{ci}^s$ | Column-integrated thermospheric temperature derived from simulated disk data |
| $T_{DISK}$ | GOLD Level 2 version 2 thermosphere temperature product |
| $T_{MSIS}$ | MSIS temperature at the altitude $z_{T_{ci}^s}$ (See Fig. 4) |
| $Q_{JH}$ | AMGeO hemispherically integrated Joule heating rate |
| $\rho_{SWARM}^{460\ km}$ | SWARM A mass density at 460 km |
| $\Sigma O/N_2^G$ | GOLD Level 2 version 2 column O/N$_2$ ratio |

**Table 1: Definitions of symbols**


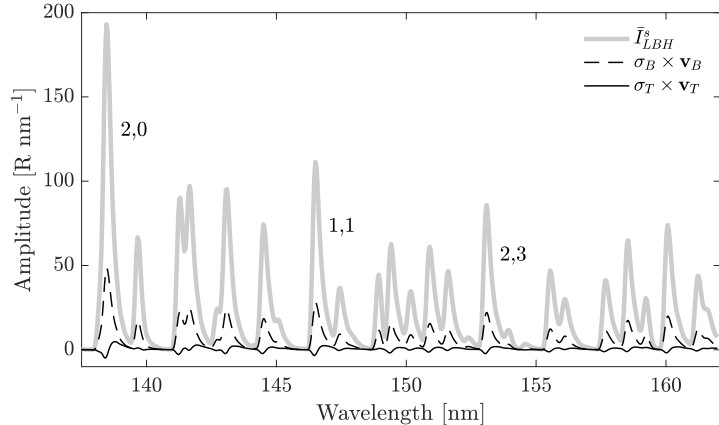

**Figure 1: Simulated top-of-atmosphere mean LBH emissions (gray), $\overline{I_{LBH}^s}$ , and the principal components associated with overall brightening of emissions (dashed black), $v_B$, and the temperature signal (solid black),**

$v_T$, **scaled by their respective eigenvalues, $\sigma_B$ and $\sigma_T$. These two principal components account for 99.9% of the variability about the mean. $\sigma_B{}^2$ accounts 98.3% of the total variability in the LBH emissions and $\sigma_T{}^2$ accounts for 1.6% of the total variability in the simulated LBH emissions over the entire disk over the period of November 2–8, 2018.**



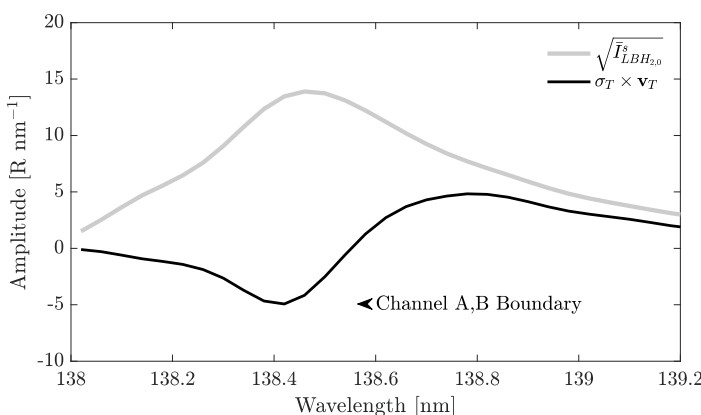


**Figure 2: Principal component associated with temperature (black line), $v_T$, over the LBH (2,0) band scaled by the eigenvalue, $\sigma_T$. $v_T$ represents an effective skewing of the emission feature to longer wavelengths or shorter wavelengths. Emissions at 135.56 nm, where $v_T$ changes the sign, are unaffected by $v_T$ (i.e., independent of temperature), and provide a boundary location to divide the (2,0) band into channels A and B.**

**The shot noise level (gray) that is the dominant random noise source is also shown.**

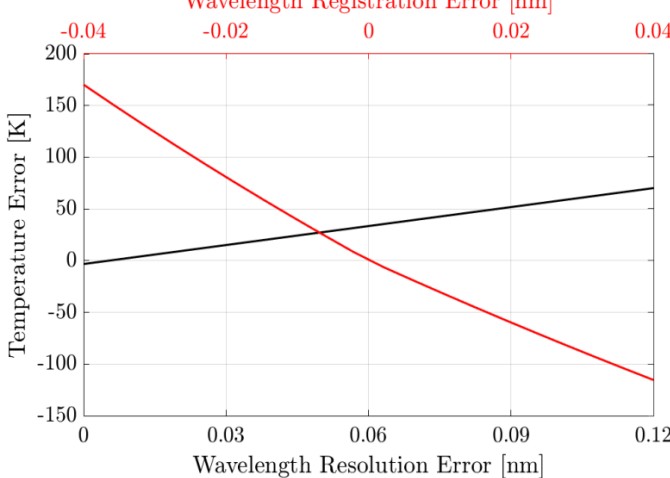

**Figure 3: Expected bias errors in $T_{ci}$ as a function of varying degrees of the two dominant known sources**

**systematic measurement errors: the instrument's wavelength registration (red) and variations in the wavelength resolution (black).**





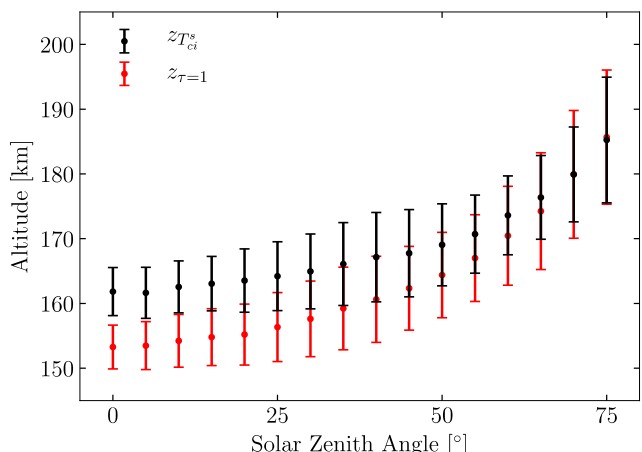

**Figure 4: The mean and standard deviation of the altitude of the simulated WAM temperature that is closest to $T_{ci}^s$, $z_{T_{ci}^s}$, as a function of SZA over the simulation period of November 2–8, 2018 (black). The mean and standard deviation of the peak of the LBH contribution function, $z_{\tau=1}$, is also shown as a function of SZA based on forward modeling of LBH disk emissions using the same WAM simulation (red).**

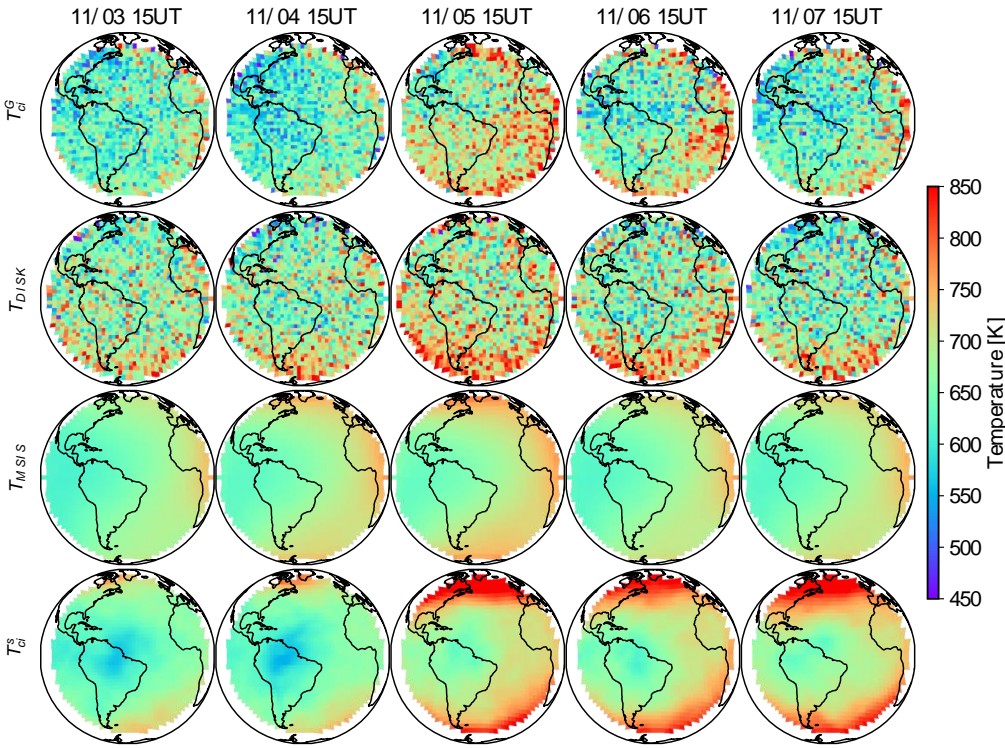


**Figure 5: Comparison of $T_{ci}^{G}$ with $T_{DISK}$, $T_{MSIS}$, and $T_{ci}^{s}$ over Earth's disk viewed by GOLD for a five-day window from November 3-7, 2018 at about 15 UT, noon LT at the center of the disk (47.5ºW, 0ºN). A small geomagnetic storm has commenced the evening of November 4th and lasted through November 5th.**



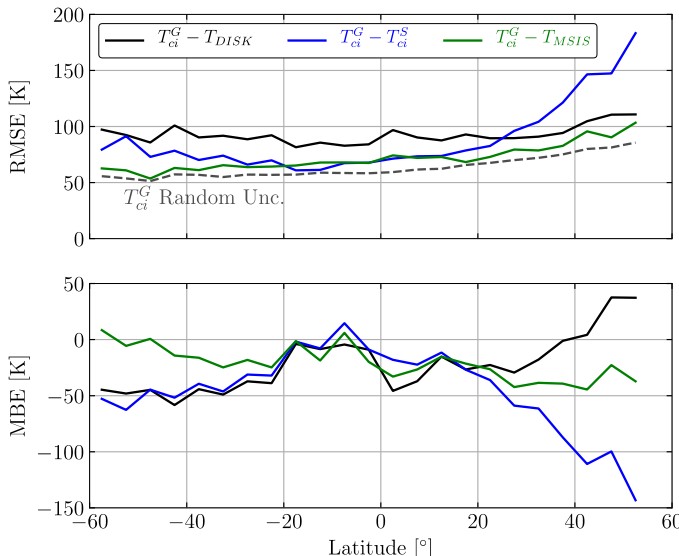


**Figure 6: Root mean squared difference (top) and mean bias difference (bottom) of $T_{ci}^G$ from $T_{DISK}$, $T_{MSIS}$, and $T_{ci}^s$ computed for a given latitude over the period of November 2–8, 2018. The random uncertainty in $T_{ci}^G$ is also shown by the gray dashed line (top).**

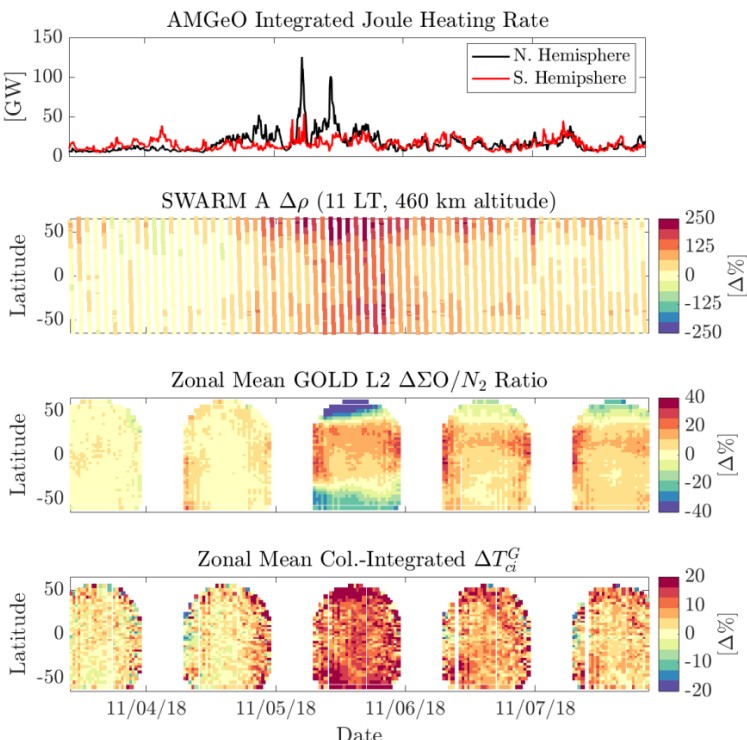

**Figure 7: Storm-time response of the thermosphere as observed in three thermospheric variables as well as the hemispherically integrated Joule heating estimated using AMGeO. The percent change in mass density, ($\Delta\rho_{SWARM}^{460\,km}$), column O/N$_2$ ratio ($\Delta\Sigma O/N_2$), column-integrated temperature ($\Delta T_{ci}^G$) are computed with quiet-time conditions on November 2nd, 2018.**
