# Peer review of "Deriving column-integrated thermospheric temperature with the $N_2$ Lyman-Birge-Hopfield (2,0) band"

_Atmospheric Measurement Techniques, 2021_

## Author Comment (AC1)

**We thank the reviewer for their thoughtful comments. Our responses to each of the major and minor comments are provided below. The text in normal font is our direct response to the reviewer and the text in italic font is the text that will be added/edited in the manuscript.**

**Major Comments:**

1. **Lookup Table (Lines 161-162): The lookup-table data has not been provided anywhere. I would suggest including them as a part of publicly available data, so that someone interested in reproducing/verification by independent means can use/validate them.**

   The lookup table will be provided as part of the publicly available data for reproducibility and verification. Note that the lookup table is dependent on instrument performance so the provided table will be most appropriately used for the November 2-8, 2018 period where we have quantified GOLD's performance (wavelength resolution and registration variations along the detector). We cannot guarantee accurate temperatures outside of this period.

2. **PCA of simulated LBH emissions: Line 115: "The second leading….(.. explained later)". This is not clear to me how the 2nd leading mode would only contain the temperature variability. Why would it not contain, for example, the geomagnetic. variability? Can you use a bunch of simulated spectra corresponding to temperatures in the 300-1500 Kelvin range and show that the 2nd leading mode is associated only with temperature changes?**

   The modes of variability derived from data via PCA decompose the variability in data into orthogonal directions that are not necessarily associated with a particular geophysical source of variability. The shape of the second mode suggests that it is capturing the broadening of individual LBH bands that can only be attributed to changes in the rotational temperature of $N_2$. Using the associated coefficients to the modes of variability, we can investigate how the variability in data at a specific time and location can be projected into each mode to gain insight into the source (geomagnetic activity, SZA, OZA, etc.).

   In the case of increased geomagnetic activity, for example, we see coefficients associated with the first mode of variability increase particularly in the high latitudes as there is more excitation of LBH emissions. At the same time, we see the coefficients associated with second mode increase as temperatures rise and the LBH bands broaden.

The following text will be added to the beginning of Section 2.2 (line 103) to help clarify the meaning of the PCA results.

*"PCA is a data reduction technique that is useful for identifying the dominant orthogonal modes of variability from data. PCA is applied here using eigenvalue decomposition of a sample covariance matrix, $\mathbf{S}_{\lambda\lambda}$, of simulated LBH emissions, $I^S_{LBH}$, at wavelengths, $\lambda$, is computed from aggregated data sets of simulated emissions of the LBH band system during 2–8 November 2018 for a total of N = 8.1×10⁴ samples.*

$$S_{\lambda\lambda} = \frac{1}{N-1} \sum_{i=1}^{N} I^{S'}_{LBH_i}{}^{T} I^{S'}_{LBH_i}$$

$$I^{S'}_{LBH_i} = I^S_{LBH_i} - \overline{I^S_{LBH}}$$

*$\overline{I^S_{LBH}}$ is the mean LBH spectrum of the N samples. The useful results of PCA for this investigation are a set of eigenvectors (principal components), $\mathbf{v}$, that describe the mode of variability in the LBH band system, with associated eigenvalues, $\sigma$. Suppose that $\mathbf{v}$ is an orthonormal set of spatiotemporally invariant basis and spatiotemporal dependent coefficients, $\mathbf{c}$, represent the amplitude of the mode for each disk emission sample at a given time, $t_i$, and location, $r_i$, then $I^{S'}_{LBH}$ can be expressed:*

$$I^{S'}_{LBH_i}(\lambda, r_i, t) = c_1(r_i, t_i)\,\mathbf{v}_1(\lambda) + c_2(r_i, t_i)\,\mathbf{v}_2(\lambda) + \ldots + c_n(r_i, t_i)\,\mathbf{v}_n(\lambda) + \mathbf{d}'(\lambda, r_i, t_i)$$

*where $\mathbf{d}'(\lambda, \mathbf{r}, t)$ is the residual after subtracting the mean and the sum of n weighted modes from $I^S_{LBH_i}$. The total variance of $\mathbf{c}$ matches $\sigma^2$ for that mode.* "

In addition to this added text, Figure 2 will be updated to further illustrate the meaning of the second mode of variability.

[Figure]

*Figure 2: The second principal component (black line), $v_T$, over the LBH (2,0) band and the normalized amplitude of the LBH (2,0) band at six $N_2$ rotational temperatures, $T_r$. Emissions at 138.56 nm, where $v_T$ changes the sign, are independent of temperature, and provide a boundary location to divide the (2,0) band into channels A and B.*

3. **Line 126: Shot noise: It is said that the spectra are just simulated/model/synthetic spectra. How can a model/simulated spectra will contain shot noise? Are you using a set of spectra or introducing some random noise in the spectra and then calculating the shot noise? Please add more explanations.**

   The reviewer makes a good point. In defining the shot noise amplitude in Section 2.2 to compare against the second mode, we simply took the square root of the mean brightness in Rayleighs of each spectral bin of the (2,0) band. This is incorrect as shot noise is instrument specific and should be run through an instrument simulator. In addition to this point, we have determined that it is not appropriate to use the principal component analysis results to quantify signal-to-noise ratio as the modes of variability and associated coefficients do not provide total signal amplitude at a given time and location, only deviations in signal amplitude from the mean.

   For these reasons, we will remove all text associated with quantifying the temperature signal-to-noise in Section 2.2. We have also removed the shot noise amplitude in Figure 2 (shown above). Removing this text does not change the major results of this manuscript.

4. **Lines 243-246: Variation in wavelength registration: Better used an atomic line but try to avoid OI-135.6 nm as it is very strong emission and on occasions degrades the detector. Variation in wavelength resolution: Again, better try to use some atomic line other than OI-135.6 nm.**

   With the understanding that the GOLD team is using atomic lines for both wavelength registration and resolution estimates, we argue that while wavelength resolution estimates likely need an atomic line to prevent the rotational structure of a molecular band interfering with the estimate of the width of the feature, estimates of wavelength registration do not need an atomic line. This is because wavelength registration is estimated with the location of the peak of the band (in this case (2,0) band) where this peak does not vary with the rotational structure of the band. However, this peak does vary with the wavelength resolution, so the resolution must first be estimated before fitting

the (2,0) band to estimate the registration. This procedure is used in the manuscript.

The text in line 243-246 will be updated as follows:

"*Variations in wavelength resolution along the GOLD detector are identified with the FWHM of the OI 135.6 doublet through fitting a 2-gaussian distribution. Variations in the wavelength registration are identified by differencing the modeled peak wavelength given the fitted OI 135.6 doublet FWHM by the peak wavelength determined by fitting a log-normal distribution to the (2,0) band. Note that the degradation of the detector due to the strength of the OI 135.6 doublet can cause errors in the spectral resolution estimate, but significant degradation had not occurred by 2-8 November 2018.*"

5. **Line 183-185: GOLD case study: Why you are not using the errors available in the L1C data? Why do you need to simulate the error?**

   We are using the errors in photon counts provided in the L1C data to simulate errors in temperature.

   The text in line 183-185 will be updated as follows:

   "*The $T_{ci}$ random measurement error given the random error in photon counts provided in the GOLD L1C data is quantified using Monte Carlo (MC) samples of simulated $T_{ci}$ derivations considering the viewing conditions and instrument performance (McClintock et al., 2020a,b).*"

6. **Line 224: "T_{ci}^{G} is also....based on the SZA." In the previous section it is stated that sampling at peak altitudes introduces 30-90K error. Then why are you using MSIS sampled at peak altitudes. I would recommend calculating GOLD equivalent effective temperatures using MSIS profiles and contribution functions from radiative transfer model. It will give better comparison with GOLD L2-Tdisk, particularly with version 3 TDISK. This can be presented as an additional row in the comparison (Figure 5).**

   For the comparison to MSIS in Section 4, we do not sample MSIS at the peak of the contribution function, $z_{\tau=1}$, (red points in Figure 4) for the given SZA but instead at the altitude with the temperature that most closely matches the derived temperature based on simulated derivations, $z_{T_{ci}^s}$, (black points in Figure 4). This is stated in line 226-227.

While responding to the reviewer's comment, we have realized that the contribution function is not only dependent on SZA but also on OZA. The sampling of MSIS thus should consider the SZA and OZA. The following figure will be added to Section 3.3.

[Figure]

*Figure \*: Pressure at the peak of the LBH contribution function, $p_{\tau=1}$, as a function of SZA and OZA determined from forward modeling WAM simulations for the period of November 2-8, 2018 considering realistic forcing conditions. LBH emissions are on constant pressure level surfaces given the solar and observing zenith angles. Approximate corresponding altitudes in the WAM simulations are also provided but note that these altitudes will vary depending on the forcing conditions.*

Text corresponding to the new Figure \* will be added starting on line 198 in Section 3.3 as follows:

*"The peak and shape of the LBH contribution function changes with SZA and observing zenith angle (OZA). Figure \* shows the peak of the LBH contribution function given the SZA and OZA determined by forward modeling WAM simulations. The peak of the contribution function decreases in pressure (increases in altitude) for increases in SZA and OZA with a stronger dependence on SZA."*

Based on the information summarized in this new figure, we have remade Figure 4 (shown below), removing the OZA dependence and plotting in terms of pressure.

[Figure]

*Figure 4: The mean and standard deviation of the pressure for the simulated WAM temperature that is closest to $T_{ci}^s$ , $p_{T_{ci}^s}$, as a function of SZA for OZA = 40° over the simulation period of 2–8 November 2018 (black). The peak of the LBH contribution function, $p_{\tau=1}$, is shown as a function of SZA based on forward modeling of LBH disk emissions using the same WAM simulation (red). This peak is constant with respect to pressure level for a given SZA. The approximate altitudes for the pressures are also provided.*

The conclusions that can be drawn from the updated Figure 4 remain the same in that the derived temperature is a column-integrated quantity and should not be attributed to the altitude of the peak of the contribution function. However, the interpretation of Figure 4 in Section 3.3 will be updated. Updates will include changing $z_{\tau=1}$ to pressure level, $p_{\tau=1}$, and $z_{T_{ci}^s}$ to pressure level, $p_{T_{ci}^s}$. Lines 203-214 will be replaced with the following text:

*"There is a clear difference in $p_{\tau=1}$ and $p_{T_{ci}^s}$ observed in their respective dependences on SZA where the range of $p_{T_{ci}^s} = 3 \times 10^{-5}$– $5 \times 10^{-5}$ hPa and the range of $p_{\tau=1} = 2 \times 10^{-5}$– $5.5 \times 10^{-5}$ for SZAs between 5°–70°. The weaker dependence of $p_{T_{ci}^s}$ on SZA can be explained by the FWHM of the contribution function which can span 60 km at low SZA and 90 km for high SZA (Laskar et al., 2020). The contribution function acts as an averaging kernel for temperature over these large vertical widths that reduces the SZA dependence relative to $p_{\tau=1}$. The net result is derived temperatures that are generally hotter than temperatures at $p_{\tau=1}$ ($p_{T_{ci}^s} < p_{\tau=1}$) for low SZA and temperatures that are generally cooler than temperatures at $p_{\tau=1}$ ($p_{T_{ci}^s} > p_{\tau=1}$) for high SZA. Figure 4 also shows considerable variability in $p_{T_{ci}^s}$ (up to $1.5 \times 10^{-5}$ hPa or ~10 km for the simulation conditions) at a given SZA that reflects variability in the vertical*

*temperature structure within the width of the contribution function for varying forcing conditions."*

7. **Section 4: The authors used the unbinned data from an old release version-2 (V2), which I cannot locate in the two GOLD repositories provided in the data availability section. As there is poor signal to noise (SNR) concern and potential bias concern, I would suggest revising the analysis and results with version 3 (V03) GOLD TDISK and 2x2 binned L1C data (L1C-V03). Specifically, revise Figure 5 with the V03 data.**

The analyses have been revisited with version 3 of the GOLD TDISK data that use binned L1C data. Figures 5 and 6 have been updated with the V03 data (shown below). There are no major changes to results associated with updated Figures 5 and 6. Considering the V03 product, there is in general better agreement between $T^G_{ci}$ and TDISK, but a comparison between $T^G_{ci}$ and TDISK V03 as a function of SZA and OZA (see the response to Major Comment 6) shows systematic differences between these datasets as discussed as part of the response to Major Comment 8.

[Figure]

*Figure 5: Comparison of $T^G_{ci}$ with $T_{DISK}$, $T_{MSIS}$, and $T^S_{ci}$ over Earth's disk viewed by GOLD for a five-day window from 3-7 November 2018 at about 15 UT, noon LT at the center of the disk (47.5ºW, 0ºN). A small geomagnetic storm has commenced the evening of 4 November and lasted through 5 November.*

[Figure]

*Figure 6: Root mean squared difference (top) and mean bias difference (bottom) of $T_{ci}^G$ from $T_{DISK}$, $T_{MSIS}$, and $T_{ci}^s$ computed for a given latitude over the period of 2–8 November 2018. The random error in $T_{ci}^G$ is also shown by the gray dashed line (top).*

8. **Lines 277: Previously the authors mentioned that this retrieval is unaffected by biases in emission intensities as the absolute values are not important, but the spectral shapes are. Then why would systematic errors in intensities, which will basically introduce some bias, would introduce bias in temperature calculation? This also contradict the conclusions in lines 318-322, which says absolute band intensities are not required.**

The systematic errors referred to in line 277 arise from wavelength registration and resolution errors which changes the relative magnitude of each channel and ultimately the resulting temperature. We have been able to significantly (although not completely) reduce the systematic errors in temperature arising from GOLD wavelength registration and resolution errors using the procedure described in Major Comment 4.

This procedure only requires relative magnitudes in each channel and not radiometrically calibrated absolute intensities. This is a major motivation for the procedure that considerably simplifies the forward model (and associated errors) from a forward model that would need an airglow volume emission rate model like GLOW and a radiative transfer model to determine absolute intensities to a forward model that only consists of the LBH rotational vibrational

band model to determine relative magnitudes in each channel for a given temperature.

Further comparisons between $T^G_{ci}$ and TDISK were performed to assess biases in the temperatures that are attributable to differences in the retrieval techniques. The following new figure and text will be added to Section 4.2 as follows.

[Figure]

*Figure \*\*: Mean $T^G_{ci}$ and $T_{DISK}$ temperatures as a function of SZA and OZA for the period of November 2-8, 2018 considering 5º binning in SZA and OZA.*

*"Fig. \*\* shows that $T^G_{ci}$ and $T_{DISK}$ have different dependencies on viewing conditions. $T^G_{ci}$ increases with both SZA and OZA while $T_{DISK}$ increases with OZA but remains relatively flat with SZA even decreasing for SZA > 25º. There are two likely explanations for these apparent dependencies:*

1. *The derived temperatures reflect real thermospheric changes with viewing conditions because the contribution function is peaking at different pressures as shown in Figure \*.*
2. *The derived temperatures reflect biases with viewing conditions because the LBH emission intensity is changing. For SZA, intensity decreases with increasing angle due to reduced LBH excitation. For OZA, intensity increases with increasing angle due to the increased airmass in the line of sight.*

*To test which hypothesis best describes the dependence of $T^G_{ci}$ and $T_{DISK}$ on viewing conditions, Figure \*\* is correlated to the pressure at the peak of the LBH contribution function, $p_{\tau=1}$, (Figure \*) and to the mean LBH intensity measured by GOLD over the same period as a function of SZA and OZA. $T_{DISK}$ is uncorrelated (r=-0.08) with $p_{\tau=1}$*

*and strongly correlated (r=0.75) with LBH intensity. In contrast, $T^G{}_{ci}$ is strongly correlated (r=-0.84) with $p_{\tau=1}$ and weakly correlated (r=-0.28) with LBH intensity. The correlations suggest that $T^G{}_{ci}$ is in agreement with apriori knowledge of the changes in $p_{\tau=1}$ with viewing conditions and is less susceptible to biases from LBH intensity changes with viewing conditions. This result supports the claim that $T^G{}_{ci}$ reduces bias by only using relative magnitudes between channels and not absolute radiometrically calibrated LBH intensities."*

**Minor comments:**

**Line 107: The sentence may be revised for clarity.**

    See reply to Major Comment 2

**Line 180-181: Provide reference?**

    This is determined by our own calculations based on the difference of the mean O2 absorption cross sections in channels A and B defined in the manuscript and constraints on column density of O2 along the line of sight. After revisiting these calculations, the O2 affect is much smaller than the stated 1.5%. The percent difference in the mean O2 absorption cross section between the two channels is 1.5%. This corresponds to an even smaller difference in transmittance given the low column density of O2 between the emission region and instrument.

    The text in line 180-181 will be updated as follows:

*"Sources that cause relative differences in the channel intensity other than temperature, such as differences in the $O_2$ absorption cross sections, must also be considered. There is only a 1.5% difference in the mean absorption cross section between the two channels that corresponds to a negligible difference in transmittance due to $O_2$ along the line of sight."*

**Line 242: Full disk measurements goes on until 23 UTC.**

    This mistake will be updated in the revised manuscript.

**Line 288: What is the x-axis in figure 7? Is it local time at all longitudes or local times at fixed longitude?**

    The x-axis is the date/time in UT. It is not in local time. The zonal mean is computed considering all longitudes over the disk for a particular GOLD scan.

---

## Author Comment (AC3)

**Updated Figure 5:**

The $T_{MSIS}$ temperatures have been updated with altitude sampling as a function of both OZA and SZA instead of just SZA like initially performed. This updated sampling results in cooler $T_{MSIS}$ temperatures such that there is now the best agreement between $T^G_{ci}$ and $T_{DISK}$. The interpretation of Figure 5 is updated in the manuscript. This update does not affect the major conclusions in the manuscript.

---

## Author Comment (AC4)

We thank the reviewer for their thoughtful and helpful comments. Our responses to each of the specific comments are provided after the general comments are restated. The text in bold font is the reviewers' comments. The text in normal font is our direct response to the reviewers and the text in italic font is the added or modified text included in the revised manuscript.

**Specific comments**

1. **Line 90 – Is the factor of 1.6 mentioned here an issue with the current approach? My understanding form later sections is that it is not. If this is the case, I believe it would be worth explicitly stating that here.**

   The factor of 1.6 is not an issue with the current approach. This is only true because we have recently started using the vibrational population rates provided in Ajello et al. (2020) that were determined from GOLD data instead of the theoretical Franck Condon factors as stated in line 96. However, there are uncertainties around the vibrational population rates in Ajello et al. (2020) which adds another error source that will be discussed in the response to Comment 2. Section 2.1 will be updated such that the discussion on the excitation and extinction source will be removed and the focus will be on the band model as follows:

   Section 2.1: *The forward model used to produce synthetic LBH emissions is built with the Global Airglow Model (GLOW) and a radiative transfer model (Solomon, 2017). GLOW computes LBH volume emission rates as a function of altitude that are input into the radiative transfer model to produce line-of-sight emissions of the LBH band system. The most important component of the forward model for the purposes of deriving thermospheric temperatures is the LBH vibrational-rotational band model (Budzien et al., 2001). The band model is a look-up table of laboratory spectra that specifies, for a given temperature, a unique spectrum for the upper vibrational states v'=0–9 of $N_2$. In the current implementation of the forward model, the v'=0–9 vibrational population rates are those provided in Ajello et al. (2020) Figure 8 based on GOLD observations and are held constant. The population rate distribution can vary with the energy distribution of the electron flux in addition to variation in excitation sources other than direct excitation such as radiative cascade and collision-induced electronic transition (Ajello et al., 2020, Eastes et al., 2000a,b; Ajello et al., 1985). Ajello et al. 1985 states that excitation thresholding should be included in airglow models to accurately reproduce LBH band intensity. However, as discussed in the following section, absolute band intensity is not needed to extract the $N_2$ rotational temperature.*

2. **Line 177 – Is this statement true, if the model for LBH with temperature is imperfect?**

Thank you for pointing this out. No, this statement would not be true if the model for LBH with temperature is imperfect. The model for LBH with temperature is the rotational-vibrational band model. The greatest source of imperfection to this model is the specification of the v'=0-9 population rates. The manuscript in various sections is updated and a new figure is added to Section 3.2 to quantify this error as follows:

Abstract – line 12: *The benefits of the two-channel ratio approach include a reduction in representativeness error as radiometrically calibrated LBH intensities are not required in the derivation procedure and a reduction in systematic measurement error caused by variations in the instrument performance across the LBH band system as a fully resolved system is also not required.*

Section 3.2 – line 177: deleted

Section 3.2 – added text: *Sources of representativeness error are those that cause relative differences in the channel intensity other than temperature that are not captured in the rotational vibrational band model. Photoabsorption by $O_2$ is one source to consider. There is only a 1.5% difference in the mean absorption cross section between the two channels that corresponds to a negligible difference in transmittance due to $O_2$ along the line of sight considering the $O_2$ absorption cross section variation with temperature. Another source of representativeness error associated with the (2,0) band is due to the overlap of the bright (2,0) transition and the weak (5,2) transition. Inaccurate specification of the v'=2 and v'=5 vibrational population rates would cause a slight change in shape of the band with respect to the observations that would be interpreted as a change in temperature. Figure 8 in Ajello et al. (2020) provides the v'=0-6 population rates and their uncertainties. These uncertainties are used to determine the associated error in the derived temperatures using the (2,0) band due to inaccurate specification of the v'=2 and v'=5 population rates. It is important to note that this representativeness error does not exist if the (1,1) or (2,3) bands are used in the derivation instead of the (2,0) band, however, these bands are much weaker and suffer from significantly larger random error due to shot noise. Figure \*\*\* shows the total random measurement error and representativeness error in the derived temperature using the (2,0) band. The representativeness error is a function of temperature while the random measurement error is a function of the (2,0) band intensity.*

[Figure]

*Figure \*\*\*: Total random measurement error (not including particle noise) and representativeness error for $T_{ci}$ using the (2,0) band. The range of (2,0) band counts for GOLD data (250 ×250 km resolution at nadir) used in the case study in Section 4 is highlighted by the grey box.*

Section 5 – line 318: *In this two-channel ratio approach, representativeness errors originating from forward modeling are reduced because radiometrically calibrated LBH band intensities are not required in the derivation procedure, and negative impact of systematic measurement errors, stemming from variations across the band system in the instrument's wavelength registration and resolution, are reduced because a fully resolved LBH band system is not required.*

3. **Line 180 – I believe that the $O_2$ absorption cross-section also varies (albeit not strongly) as a function of temperature. This will further complicate this factor, although it is likely still minor.**

   The $O_2$ absorption cross section does vary with temperature, however, this temperature dependence does not change the relative absorption between the two channels. The main text is updated as follows:

   *There is only a 1.5% difference in the mean absorption cross section between the two channels that corresponds to a negligible difference in transmittance due to $O_2$ along the line of sight considering the $O_2$ absorption cross section variation with temperature.*

4. **Line 182 – It is certainly true that the shot noise, which is proportional to the square root of the emission signal, is a major part of the instrumental**

**noise. However, particle noise is, at least at some times, an additional random noise source. Importantly, it's behavior is not the same as the shot noise as it is unrelated to the brightness of the signal being observed. See for example the description of the particle background and its associated flag in GOLD Release Notes Revision 4 - https://gold.cs.ucf.edu/wp-content/documentation/GOLD_Release_Notes_Rev4.1.pdf. This may, potentially, be an important consideration in the case study presented in this manuscript.**

We agree with the reviewer. The need to consider particle noise as another random noise source is a strong point. We have reviewed the particle background counts and its associated flag for observations used in the case study and found relatively low counts (~0-0.3) with the high background flag set to false. Therefore, we do not think the particle background counts affect the results of this manuscript, but it will be important moving forward to (1) quantify the statistics of background counts as a function of wavelength and (2) quantify the associated temperature errors. The following text is included in Section 3.2:

*Particle background counts is at times an additional random noise source. For the case study with GOLD data, the particle backgrounds were low as indicated by the High_Background flag in the Level 1C data and therefore this error source is not considered. The statistics of background counts and the associated temperature errors should be quantified for the general application of this technique to any time period.*

**Line 267 – The east-west gradient that is described here is not clear to me in Figure 5. I would recommend that this be demonstrated more clearly, perhaps in a line-figure such as Figure 6, as I believe it is an important point that current, at least I struggle to see from the image.**

To address the reviewer's concern, Figure 6 (shown below) has been updated such that the RMSD plot has been removed and replaced with the MBD as a function of longitude. Note that the comparison between $T\_G\_ci$ and $T\_MSIS$ changed because the MSIS temperatures are updated based on new sampling with respect to both OZA and SZA (see AC3 to Reviewer 1). Also, note that since the comparison has been updated with $T\_DISK$ version 3, the interpretation of Figure 5 and Figure 6 has changed but the major conclusions from the manuscript remain unchanged. The east-west gradient is more pronounced now in the TDISK product similar to that in $T\_G\_ci$, although the $T\_G\_ci$ is still more pronounced as seen in the updated Figure 6 below. These interpretations will be updated in the manuscript.

[Figure]

*Figure 6: Mean bias difference of $T_{ci}^G$ from $T_{DISK}$, $T_{MSIS}$, and $T_{ci}^S$ as a function of latitude (left) and longitude (right) for November 2-8, 2018 at 15 UT.*

5. **Figure 5 – The range over the disk where T_ci_G appear is smaller than that of Tdisk. Is the origin of this a differences in the solar zenith angle ranges, or some other criteria used in the approach described here that differs from the publicly available Tdisk?**

   There was an error in the plotting routine that masked more T_ci_G compared to TDISK as a function of solar and observing zenith angle. This error has been corrected and each of the temperature products is plotted over the same range of observing zenith angle and solar zenith angle as shown in the updated Figure 5.

---

## Author Response (AR1)

We thank the reviewers for their thoughtful comments. Our responses to each of the major and minor comments are provided below. The text in normal font is our direct response to the reviewer and the text in italic font is the text that are added/edited in the manuscript.

**Reviewer 1 response:**

**Major Comments:**

1. Lookup Table (Lines 161-162): The lookup-table data has not been provided anywhere. I would suggest including them as a part of publicly available data, so that someone interested in reproducing/verification by independent means can use/validate them.

The lookup table is provided as part of the publicly available data for reproducibility and verification **as described in the Data Availability section**. Note that the lookup table is dependent on instrument performance so the provided table is most appropriately used for the November 2-8, 2018 period where we have quantified GOLD's performance (wavelength resolution and registration variations along the detector). We cannot guarantee accurate temperatures outside of this period.

2. PCA of simulated LBH emissions: Line 115: "The second leading....(.. explained later)". This is not clear to me how the 2nd leading mode would only contain the temperature variability. Why would it not contain, for example, the geomagnetic. variability? Can you use a bunch of simulated spectra corresponding to temperatures in the 300-1500 Kelvin range and show that the 2nd leading mode is associated only with temperature changes?

The modes of variability derived from data via PCA decompose the variability in data into orthogonal directions that are not necessarily associated with a particular geophysical source of variability. The shape of the second mode suggests that it is capturing the broadening of individual LBH bands that can only be attributed to changes in the rotational temperature of N2. Using the associated coefficients to the modes of variability, we can investigate how the variability in data at a specific time and location can be projected into each mode to gain insight into the source (geomagnetic activity, SZA, OZA, etc.).

In the case of increased geomagnetic activity, for example, we see coefficients associated with the first mode of variability increase particularly in the high latitudes as there is more excitation of LBH emissions. At the same time, we see

the coefficients associated with second mode increase as temperatures rise and the LBH bands broaden.

The following text is added to the beginning of Section 2.2 (Lines 94-107) to help clarify the meaning of the PCA results.

"PCA is a data reduction technique that is useful for identifying the dominant orthogonal modes of variability from data. PCA is applied here using eigenvalue decomposition of a sample covariance matrix,  $S_{\lambda\lambda}$ , of simulated LBH emissions,  $I_{LBH}^{s}$ , at wavelengths,  $\lambda$ , is computed from aggregated data sets of simulated emissions of the LBH band system during 2–8 November 2018 for a total of N = 8.1×104 samples.

$$S_{\lambda\lambda} = \frac{1}{N-1} \sum_{i=1}^{N} I_{LBH_i}^{S'} I_{LBH_i}^{S'}$$
$$I_{LBH_i}^{S'} = I_{LBH_i}^{S} - \overline{I_{LBH}^{S}}$$

 $\overline{I}_{LBH}^{s}$  is the mean LBH spectrum of the N samples. The useful results of PCA for this investigation are a set of eigenvectors (principal components), **v**, that describe the mode of variability in the LBH band system, with associated eigenvalues,  $\sigma$ . Suppose that **v** is an orthonormal set of spatiotemporally invariant basis and spatiotemporal dependent coefficients, **c**, represent the amplitude of the mode for each disk emission sample at a given time,  $t_i$ , and location,  $r_i$ , then  $I_{LBH}^{sr}$  can be expressed:

$$I_{LBH_{i}}^{s'}(\lambda, r_{i}, t) = c_{1}(r_{i}, t_{i}) \nu_{1}(\lambda) + c_{2}(r_{i}, t_{i}) \nu_{2}(\lambda) + \dots + c_{n}(r_{i}, t_{i}) \nu_{n}(\lambda) + d'(\lambda, r_{i}, t_{i})$$

where  $d'(\lambda, r, t)$  is the residual after subtracting the mean and the sum of n weighted modes from  $I_{LBH_i}^s$ . The total variance of **c** matches  $\sigma^2$  for that mode. "

In addition to this added text, Figure 2 is updated to further illustrate the meaning of the second mode of variability.

Figure 2: The second principal component (black line),  $v_T$ , over the LBH (2,0) band and the normalized amplitude of the LBH (2,0) band at six  $N_2$  rotational temperatures,  $T_r$ . Emissions at 138.56 nm, where  $v_T$  changes the sign, are independent of temperature, and provide a boundary location to divide the (2,0) band into channels A and B.

3. Shot noise: It is said that the spectra are just simulated/model/synthetic spectra. How can a model/simulated spectra will contain shot noise? Are you using a set of spectra or introducing some random noise in the spectra and then calculating the shot noise? Please add more explanations.

The reviewer makes a good point. In defining the shot noise amplitude in Section 2.2 to compare against the second mode, we simply took the square root of the mean brightness in Rayleighs of each spectral bin of the (2,0) band. This is incorrect as shot noise is instrument specific and should be run through an instrument simulator. In addition to this point, we have determined that it is not appropriate to use the principal component analysis results to quantify signal-to-noise ratio as the modes of variability and associated coefficients do not provide total signal amplitude at a given time and location, only deviations in signal amplitude from the mean.

For these reasons, we have removed all text associated with quantifying the temperature signal-to-noise in Section 2.2. We have also removed the shot noise amplitude in Figure 2 (shown above). Removing this text does not change the major results of this manuscript.

4. Variation in wavelength registration: Better used an atomic line but try to avoid OI-135.6 nm as it is very strong emission and on occasions degrades

**the detector. Variation in wavelength resolution: Again, better try to use some atomic line other than OI-135.6 nm.**

With the understanding that the GOLD team is using atomic lines for both wavelength registration and resolution estimates, we argue that while wavelength resolution estimates likely need an atomic line to prevent the rotational structure of a molecular band interfering with the estimate of the width of the feature, estimates of wavelength registration do not need an atomic line. This is because wavelength registration is estimated with the location of the peak of the band (in this case (2,0) band) where this peak does not vary with the rotational structure of the band. However, this peak does vary with the wavelength resolution, so the resolution must first be estimated before fitting the (2,0) band to estimate the registration. This procedure is used in the manuscript.

The text is updated as follows (Lines 247-250):

"Variations in wavelength resolution along the GOLD detector are identified with the FWHM of the OI 135.6 doublet through fitting a 2-gaussian distribution. Variations in the wavelength registration are identified by differencing the modeled peak wavelength given the fitted OI 135.6 doublet FWHM by the peak wavelength determined by fitting a log-normal distribution to the (2,0) band. Note that the degradation of the detector due to the strength of the OI 135.6 doublet can cause errors in the spectral resolution estimate, but significant degradation had not occurred by 2-8 November 2018."

**5. GOLD case study: Why you are not using the errors available in the L1C data? Why do you need to simulate the error?**

We are using the errors in photon counts provided in the L1C data to simulate errors in temperature.

The text is updated as follows (Lines 169-171):

"The Tci random measurement error given the random error in photon counts provided in the GOLD L1C data is quantified using Monte Carlo (MC) samples of simulated Tci derivations considering the viewing conditions and instrument performance (McClintock et al., 2020a,b)."

6. "T\_{ci}^{G} is also....based on the SZA." In the previous section it is stated that sampling at peak altitudes introduces 30-90K error. Then why are you using MSIS sampled at peak altitudes. I would recommend calculating GOLD equivalent effective temperatures using MSIS profiles and contribution functions from radiative transfer model. It will give better comparison with GOLD L2-Tdisk, particularly with version 3 TDISK. This can be presented as an additional row in the comparison (Figure 5).

For the comparison to MSIS in Section 4, we do not sample MSIS at the peak of the contribution function,  $p_{\tau=1}$ , (red points in Figure 6) for the given SZA but instead at the pressure with the temperature that most closely matches the derived temperature based on simulated derivations,  $p_{T_{cl}^s}$ , (black points in Figure 6).

While responding to the reviewer's comment, we realized that the contribution function is not only dependent on SZA but also on OZA. The sampling of MSIS thus should consider the SZA and OZA. The following figure is added to Section 3.3.

Figure 5: Pressure at the peak of the LBH contribution function,  $p_{\tau=1}$ , as a function of SZA and OZA determined from forward modeling WAM simulations for the period of November 2-8, 2018 considering realistic forcing conditions. LBH emissions are on constant pressure level surfaces given the solar and observing zenith angles. Approximate corresponding altitudes in the WAM simulations are also provided but note that these altitudes would vary depending on the forcing conditions.

Text corresponding to the new Figure 5 is added in Section 3.3 as follows (Lines 202-204):

"The LBH contribution function peak,  $p_{\tau=1}$ , changes with solar zenith angle (SZA) and observing zenith angle (OZA) as shown in Fig. 5.  $p_{\tau=1}$  decreases in pressure (increases in altitude) for increases in SZA and OZA with a stronger dependence on SZA."

Based on the information summarized in this new figure, we have remade Figure 4 (now Figure 6), removing the OZA dependence and plotting in terms of pressure.

---

## Referee Report (RR1)

A review of 'Deriving column-integrated thermospheric temperature with the N2 Lyman–Birge– Hopfield (2,0) band', by C. Cantrall and T. Matuso

**General comments**

- The technique for determining temperatures from disk LBH emissions that is described in this manuscript represents an important new tool that may enable both further analysis of data from GOLD, but also potentially future instrumentation. The method has several strengths over existing techniques, including not requiring knowledge of the absolute brightness of the emission, not requiring a broad portion of the LBH bands to be sampled, and not requiring the kind of spectral resolution that has underpinned some other techniques. As such, this work should be of great interest to those interested in thermospheric observations, and techniques for analyzing such observations.
- The manuscript includes a good description of the uncertainties, related to instrument wavelength and noise both shot noise and particle noise.
- The detailed description in Section 3 of how the disk temperature should be interpreted as column temperatures is particularly important and as these and similar data utilized by a broader community this kind of consideration is essential.
- The particular case study, utilizing data from multiple spacecraft and centered around a moderate geogmagnetic storm provides a good demonstration of how the temperatures retrieved from the technique introduced here vary under such conditions, and demonstrates their utility to the broader scientific community.

Comments & Errors:

None noted.

---

## Author Response (AR2)

**We thank the reviewer for their thoughtful comments. Our responses to each of the minor comments are provided below. The text in normal font is our direct response to the reviewer and the text in italic font is the text that are added/edited in the manuscript.**

**Reviewer 1 response:**

**Minor Comments:**

1. **Line 41: Does "N 149.3 nm" refer to atomic oxygen nitrogen line? If so, define it. Atomic nitrogen 149.3 nm line is also referred to as N I 149.3 line.**

   Yes, "N 149.3 nm" refers to the atomic nitrogen line at 149.3 nm. The text is updated to *"N I 149.3 nm line"*.

2. **Line 143: Add more description on the procedure. It is not clear what is being done in this step (Step 2).**

   The procedure is updated as follows:

   *The procedure to determine $T_{ci}$ using the two-channel ratio consists of four steps as follows:*

   1. *Generate a set of synthetic LBH (2,0) bands at the instrument's pixel size for a range of temperature using the vibrational-rotational band model (Budzien et al., 2001).*
   2. *Apply an instrument model on each synthetic band to account for the instrument's wavelength resolution and wavelength registration.*
   3. *Bin each band into channels A and B and least squares fit the ratio, $B/A$, to temperature.*
   4. *Compute the ratio, $B/A$, from the observed LBH (2,0) band and determine $T_{ci}$ by regressing observed ratio on the predetermined relationship between the ratio and temperature.*

3. **Line 169-171: This line may need rewording, appears confusing to me.**

   The text is updated as follows:

   *"The $T_{ci}$ random measurement error due to shot noise is quantified using Monte Carlo samples of simulated $T_{ci}$ derivations considering the instrument performance (McClintock et al., 2020a,b)."*

4. **202-203: LBH contribution function (CF): Figure 5 is very interesting, which shows pressure at the peak of the CF. It would be also interesting to see how the pressure/altitude profile of CF varies with SZA and OZA. If allowed by journal length limit, I would suggest adding another figure on that, at least for some sample SZAs and OZAs.**

   We appreciate the reviewer's suggestion but after consideration we do not think there is enough information gained from the suggested figure relative to Figure 5 and also relative to figures provided in Laskar et al, 2020 and Zhang et. al. 2019 to include an additional figure in this manuscript.

5. **260-262: It is still not clear to me why the $T_{ci}$ is compared with $T_{MSIS}$. As I was suggesting earlier, better compare it with an equivalent height integrated quantity. Except $T_{MSIS}$, all other quantities are vertically integrated in Figure 7. If there is a special motivation behind showing the non-height integrated $T_{MSIS}$ then it should be made clear.**

   $T_{MSIS}$ is updated to be a height integrated quantity with the same procedure taken for $T^s_{ci}$. Figures 7 and 8 are updated along with the description of $T_{MSIS}$ in Section 4 and Table 1. There are no major changes to the results based on this update but $T_{MSIS}$ is in slightly better agreement with the other datasets for the period of interest.

6. **GOLD TDISK retrieval algorithm is not optimized for auroral latitudes (see Laskar et al., 2021 or GOLD TDISK documentation) due to very different chemical environment. Please provide a comment on the performance of your retrieval at auroral latitudes. Can it explain some of the high-latitude dissimilarities seen in Figure 7?**

   A detailed analysis of the retrieval at auroral latitudes has not been conducted. We expect that the performance will remain similar to low latitudes due to the point made in the text that the forward model only consists of a vibrational rotational band model and thus the retrieval is not affected by various excitation and extinction processes along the line-of-sight. However, if there is a change in the v'=2 and v'=5 population rates at auroral latitudes from what Ajello et al. 2020 found then the retrieval could be biased. It is not clear from Laskar et al. 2021 or the GOLD TDISK documentation why the different chemical environment at auroral latitudes affects the TDISK retrieval. The GOLD TDISK documentation suggests that the population rates appear to remain constant with respect to viewing conditions and geomagnetic conditions. We expect our retrieval performance to be unaffected as a result.